# Effects of Exercise Combined with a Healthy Diet or *Calanus finmarchicus* Oil Supplementation on Body Composition and Metabolic Markers—A Pilot Study

**DOI:** 10.3390/nu12072139

**Published:** 2020-07-18

**Authors:** Paulina Wasserfurth, Josefine Nebl, Jan Philipp Schuchardt, Mattea Müller, Tim Konstantin Boßlau, Karsten Krüger, Andreas Hahn

**Affiliations:** 1Faculty of Natural Sciences, Institute of Food Science and Human Nutrition, Leibniz University Hannover, 30167 Hannover, Germany; wasserfurth@nutrition.uni-hannover.de (P.W.); nebl@nutrition.uni-hannover.de (J.N.); schuchardt@nutrition.uni-hannover.de (J.P.S.); mueller@nutrition.uni-hannover.de (M.M.); 2Department of Exercise Physiology and Sports Therapy, Institute of Sports Science, Justus-Liebig-University Giessen, 35394 Giessen, Germany; tim.k.bosslau@med.uni-giessen.de (T.K.B.); karsten.krueger@sport.uni-giessen.de (K.K.)

**Keywords:** aging, exercise, obesity, omega-3 fatty acid, body composition, fat loss, glucose metabolism

## Abstract

Aging is accompanied by a progressive decline in muscle mass and an increase in fat mass, which are detrimental changes associated with the development of health conditions such as type-2 diabetes mellitus or chronic low-grade inflammation. Although both exercise as well as nutritional interventions are known to be beneficial in counteracting those age-related changes, data to which extent untrained elderly people may benefit is still sparse. Therefore, a randomized, controlled, 12-week interventional trial was conducted in which 134 healthy untrained participants (96 women and 38 men, age 59.4 ± 5.6 years, body mass index (BMI) 28.4 ± 5.8 kg/m^2^) were allocated to one of four study groups: (1) control group with no intervention (CON); (2) 2×/week aerobic and resistance training only (EX); (3) exercise routine combined with dietary counseling in accordance with the guidelines of the German Nutrition Society (EXDC); (4) exercise routine combined with intake of 2 g/day oil from *Calanus finmarchicus* (EXCO). Body composition (bioelectrical impedance analysis), as well as markers of glucose metabolism and blood lipids, were analyzed at the beginning and the end of the study. The highest decreases in body fat were observed within the EXCO group (−1.70 ± 2.45 kg, *p* < 0.001), and the EXDC (−1.41 ± 2.13 kg, *p* = 0.008) group. Markers of glucose metabolism and blood lipids remained unchanged in all groups. Taken together results of this pilot study suggest that a combination of moderate exercise and intake of oil from *Calanus finmarchicus* or a healthy diet may promote fat loss in elderly untrained overweight participants.

## 1. Introduction

Aging is accompanied by changes in body composition, which include a progressive decline in muscle mass as well as an increase in body fat mass [1]. If not counteracted, those changes may have multiple health consequences such as development of obesity, type-2 diabetes, chronic low-grade inflammation, or cardiovascular diseases [1,2,3]. Regular exercise is known to prevent, attenuate and even reverse the described changes [4,5,6] as it helps to prevent muscle loss, build new muscle mass, and reduce adipocyte cell size and lipid content [7]. Moreover, it has broad beneficial effects on human energy metabolism and glucose control as it elicits contraction-induced, insulin-independent glucose uptake in skeletal muscle [8,9].

However, body composition and metabolic markers are also largely influenced by the diet. A diet that is high in energy and high glycemic foods may facilitate the development of obesity, cardiovascular diseases or diabetes [10,11]. Conversely, a nutrient-dense diet providing adequate intake of macro- and micronutrients, fiber, and phytochemicals can help to prevent those diseases and support the beneficial effects of exercise [12,13,14,15,16]. To promote such “healthy diet”, the German Nutrition Society recommends a diet rich in vegetables, fruits, whole grain cereal products, daily intake of dairy products, moderate meat intake and 1–2 portions of fish per week [17].

On the other hand, there are also individual nutrients, which have been shown to positively influence body composition as well as metabolic markers. In that regard, results from several studies demonstrated that supplementation of omega-3 polyunsaturated fatty acids (n-3 PUFAs) potentially reduces fasting insulin levels in diabetic individuals and positively influences body composition by supporting body weight loss [18,19,20,21]. Moreover, n-3 PUFAs are also linked to increases in muscle mass and muscle function in healthy older adults [20]. One novel source of n-3 PUFAs that showed promising effects on the reduction of abdominal fat and glucose control in animal studies is the oil from the copepod *Calanus finmarchicus* [22,23]. Of all fatty acids present in Calanus oil (CO) >80% are bound as wax esters. In addition to providing n-3 PUFAs such as stearidonic acid (SDA, C18:4n3), eicosapentaenoic acid (EPA, C20:5n3) and docosahexaenoic acid (DHA, 22:6n3), it also contains plant sterols and astaxanthin, therefore differing from conventional marine oils. Carotenoids such as astaxanthin are also discussed to play a role in counteracting obesity [24].

Although synergistic effects of exercise combined with nutritional measures are presumed, only a few studies have investigated the effects of a combined exercise and nutritional intervention. Moreover, most of those studies were focused on the pre-diabetic, diabetic, or obese populations leaving out the elderly untrained but otherwise healthy population [25,26,27]. Further, no studies have examined whether and how exercise combined with either a healthy diet versus supplementation of CO may differ.

Given the information above, we hypothesized that moderate exercise combined with nutritional measures may result in more favorable changes in body composition and metabolic markers than exercise alone. Further, we aimed to identify whether one of the two nutritional interventions would result in greater changes than the other. Therefore, a pilot study was conducted to investigate to which extent elderly untrained, but otherwise healthy, subjects may benefit in terms of body composition, glucose control, and blood lipids when performing exercise only or exercise combined with either a general dietary recommendation or intake of CO.

## 2. Materials and Methods

### 2.1. Study Participants

In total, 134 men and women were recruited via advertisements in local newspapers and public notice boards from the general population in Hannover, Germany between August 2018 and March 2019. The main inclusion criteria for participation were: age ≥ 50 and ≤ 70 years, no exercise training aside from the daily activities for at least 2 years, a stable body weight (± 5 kg) for at least 6 months, ability to physically perform the exercise intervention (exercise capacity) and consumption of an omnivorous diet. Exclusion criteria were defined as suspicion and diagnosis of cardiovascular diseases (angina pectoris, myocardial infarction, stroke, peripheral arterial occlusive disease, heart failure, cardiac arrhythmia), type 1 and 2 diabetes, renal insufficiency and liver diseases, blood coagulation disorders, chronic gastrointestinal disorders (e.g., ulcers, Crohn’s disease), pancreatic insufficiency, immunological diseases (e.g., autoimmune diseases), intake of immunosuppressive drugs or laxatives, intake of supplements containing n-3 PUFAs, alcohol, drug and/or medicine dependency, pregnancy or lactation, retraction of the consent by the subject, concurrent participation in another clinical study, and participation in a study in the last 30 days. Inclusion and exclusion criteria were assessed using a structured screening questionnaire. Cardiovascular health was determined by resting and exercise electrocardiogram, implemented by trained professionals and a physician.

### 2.2. Study Design

This single-center, randomized controlled trial in parallel group design was conducted by trained professionals using standardized methods at the Institute of Food Science and Human Nutrition, Leibniz University Hannover, Germany. The study involved a screening phase and 12-week intervention phase with two examination days; one at the beginning (t_0_) and one at the end of the 12-week intervention (t_12_). Additionally, there was a questionnaire-based examination after six weeks (t_6_).

Ethical approval was provided by the Ethics Commission of the Medical Chamber of Lower Saxony (Hannover, Germany). In accordance with the guidelines of the Declaration of Helsinki, written informed consent was obtained from all participants prior to their participation in the study. This study is registered in the German Clinical Trial Register (DRKS00014322). The participants were randomly assigned by an independent researcher using stratified randomization according to the covariates sex, BMI, age (in descending order) to one of the four study groups: (1) control group (CON), (2) exercise only group (EX), (3) exercise and dietary counseling group (EXDC), (4) exercise and CO supplementation group (EXCO). The CON group served as control and participants of this group were asked to maintain their habitual diet and physical activity level throughout the 12-week investigation period. The EX, EXDC, and EXCO groups were instructed to perform exercise training twice a week. Participants randomized to the EX group were asked to maintain their normal diet. Participants randomized to the EXDC group were asked to adapt their diet in accordance with the dietary guidelines of the German Nutrition Society [17]. Therefore, participants of this group received an individualized nutrition counseling session by a professional nutritionist at the first examination. In general, dietary recommendations included the following advice: intake of 3 portions of vegetables and 2 portions of fruits daily, consumption of cereal products with a focus on whole-grain products, daily consumption of dairy products such as milk or cheese, limited meat intake of 300–600 g per week, consumption of fish once or twice a week, limited intake of salt and sugar [17]. In addition to exercise, participants randomized to the EXCO group were instructed to consume four capsules containing oil from *Calanus finmarchicus* (Calanus AS, Tromsø, Norway) daily during the 12-week intervention period. Therefore, participants received a counted number of capsules and were advised to return leftover capsules at the end of the study. The leftover capsules were then counted to assess the compliance in taking the supplement (at least 90% of the capsules had to be consumed). Four capsules provided 2 g of CO. The composition of the oil is shown in Table 1. Further, participants were asked to maintain their normal diet. Compliance of participants and adherence to the respective instructions was monitored within all study groups via fortnightly phone calls.

### 2.3. Exercise Training

The exercise training was performed in cooperating fitness centers after thorough instructions from a professional trainer. Each training session consisted of an initial warm-up, followed by two passes of a strength-endurance circuit.

The strength training consisted of six machine-supported exercises that included all major muscle groups and were performed for one minute each. During the initial training session, a maximum force test with three attempts was performed. The best of the three tests was scored and used to define the exercise intensity at about 60% of the participant’s maximum force for the first two weeks of training. For the subsequent six weeks, the load was increased by 10% and again by 5% for the last four weeks. The endurance exercise consisted of a four-minute bout performed on bicycle ergometers and cross-trainers at a perceived exertion corresponding to an intensity of about 15 on the Borg-Scale. In between each exercise, the participants had 30 s of rest. Including the warm-up and rest periods, the training session could be completed in approximately one hour. Compliance of the participants was assessed via a training log and a questionnaire at the end of the study.

### 2.4. Monitoring of Dietary Intake and Physical Activity

The level of regular physical activity outside of the intervention was assessed using the German Freiburger Questionnaire for Physical Activity at the beginning, after six weeks, and at the end of the study [28]. Dietary intake of the participants was monitored via 3-day dietary records at the beginning, after six weeks, and at the end of the study. The records were checked by nutritionists for completeness, readability, and plausibility. If necessary, ambiguities were clarified with the participants. Energy and nutrient intake were estimated using the software PRODI6.4 (Nutri-Science GmbH, Freiburg, Germany). Additionally, consumption of specific food groups was assessed with the food frequency questionnaire (FFQ) from the German Health Examination Survey for Adults (Studie zur Gesundheit Erwachsener in Deutschland, DEGS) of the Robert Koch Institute.

### 2.5. Bodyweight and Body Composition

All measurements were performed under standardized conditions on both days of examination at the beginning and end of the study. Participants arrived at the institute between 6:00–10 a.m., rested, and after an overnight fast (≥10 h). For the second examination, approximately the same time of day was scheduled as for the first day of examination. At first height was measured with a stadiometer (seca GmbH & Co. KG, Hamburg, Germany). Therefore, participants were advised to take off their shoes, stand with the back to the stadiometer and touch the bar with the back of their head, back and buttocks and back of their feet. Next, their bodyweight was measured lightly clothed and without shoes on a digital scale to the nearest of 0.1 kg (seca GmbH & Co. KG, Hamburg, Germany). Body composition was analyzed to a nearest of 0.1 kg using a bipolar bioelectrical impedance analyzer (BIA) (Nutriguard M, Data Input Company, Darmstadt, Germany) and the software NutriPlus 5.4.1 (Data Input Company, Darmstadt, Germany). For the measurements, the participants were instructed to lay down on a stretcher and rest for a few minutes (~5 min) to ensure a balanced distribution of body fluids before the measurement. During the measurement, participants were instructed to lay still and in a relaxed position with the limbs slightly bent from the torso. All measurements were carried out by trained nutritionists. To avoid biases due to changing personal, the same nutritionists conducted the measurements throughout the study after thorough instructions.

### 2.6. Blood Sampling and Biochemical Indices Measurement

Blood samples were drawn from the participants after an overnight fast (≥10 h) by venepuncture of an arm vein using EDTA, serum and NaF Glucose tubes (Sarstedt AG & Co. KG, Nümbrecht, Germany). All samples were stored at ~5 °C and transferred to an accredited and certified laboratory (Laborärztliche Arbeitsgemeinschaft für Diagnostik und Rationalisierung e.V., Hannover, Germany), where all analyses were performed.

Triglycerides, low-density lipoprotein cholesterol (LDL) and high-density lipoprotein cholesterol (HDL) were analyzed by a photometric method (Beckman Coulter GmbH, Krefeld, Germany). Total cholesterol and LDL/HDL-ratio was calculated from LDL and HDL values.

Fasting glucose was determined by a photometric method (Beckman Coulter GmbH, Krefeld, Germany). HbA_1c_ was analyzed using high-pressure liquid chromatography (HPLC) (Bio-Rad Laboratories GmbH, Feldkirchen, Germany). The electrochemiluminescence immunoassay method (ECLIA) using cobas 801e (Roche Diagnostics GmbH, Mannheim, Germany) was applied to determine insulin concentrations. To evaluate insulin resistance, the homeostatic model assessment (HOMA) was calculated as follows: HOMA-Index = fasting insulin (µU/mL) × fasting blood glucose (mg/dL)/405 [29].

### 2.7. Statistical Analyses

The sample size of *n* = 25 per group was based on an alpha of 0.05 and 0.80 beta to detect between-group differences, assuming an effect size of more than 0.8. With an estimated dropout rate of 15%, a total of at least 30 participants per intervention study group were recruited. Distribution of all data was assessed using a Shapiro-Wilk Test and Gaussian distribution. To assess differences between groups at baseline, data from all participants was analyzed. Because all data was not normally distributed, intergroup comparisons were assessed with Kruskal-Wallis-test while nominal data was analyzed using Chi-square-test. For further analysis, data from all participants who took part in both examination days was used. If the assumption of normality was not met, data was transformed using log, square root, or reciprocal transformation. The intervention effect on all outcomes was analyzed using a two-way repeated measures analysis of variance (ANOVA) with time (t_0_ and t_12_) and intervention (CON, EX, EXCO, EXDC). If significant time*intervention effects were detected, a post hoc analysis with Bonferroni correction was performed. *p*-values of <0.05 were considered as significant. All statistical analyses were carried out using SPSS software (version 23.0; SPSS Inc., Chicago, IL, USA).

## 3. Results

### 3.1. Baseline Characteristics

After screening for eligibility, 134 participants met the eligibility criteria and were randomly assigned to one of the four study groups (Figure 1). Of all participants randomized, 72% were female and 28% were male with a mean age of 59.2 ± 5.60 years. With a mean weight of 83.0 ± 20.2 kg and an average BMI of 28.4 ± 5.80 kg/m^2^, the study population can be classified as overweight.

At baseline, there were no differences in age, weight or BMI among the study groups (Table 2).

### 3.2. Physical Activity and Training Sessions

In compliance with the given instructions, there were no significant changes in physical activity outside the intervention among the four study groups during the study period (Appendix A). However, an overall increase in basal activity could be observed in all groups, which was mostly due to increased seasonal activities such as gardening. Additionally, no significant difference in the number of completed training sessions was detected between the EX, EXCO and EXDC group (data not shown).

### 3.3. Dietary Intake

Intake of specific food groups assessed by FFQs of all study groups before and after the intervention is presented in Appendix A. In compliance with the received dietary recommendations, participants in the EXDC group had a significant increase in fruit portions per day (*p* = 0.001) and vegetable portions per day (*p* = 0.006) as well as fish intake per week (*p* < 0.001) as compared to the other groups. Except for a significant decline of vegetable portions per day in the CON group (*p* = 0.018), no other significant changes were observed.

Dietary intake from the background diet calculated from 3-day dietary food logs is shown in Table 3. In all three exercise groups (EX, EXCO, and EXDC), average energy intake decreased about 200 kcal per day after the intervention period, yet this was not statistically different between groups. While protein intake stayed the same throughout the intervention and between groups, significant differences among all four study groups were found in fat (*p* = 0.016), carbohydrate (*p* = 0.015) and fiber intake (*p* = 0.004) after the intervention. Relative fat intake was significantly higher after 12 weeks in the CON group (+4.32 ± 8.17%E, *p* = 0.008), while no significant differences were found in the intervention groups. Similarly, CON showed a significant decrease in carbohydrate intake (−5.11 ± 6.65%E, *p* = 0.002). A significant decrease in carbohydrate consumption was also detected in the EX group (−2.76 ± 8.36%E, *p* = 0.034). Fiber intake increased significantly in the EXDC group (+3.00 ± 8.39 g/day, *p* = 0.021) while it decreased in the EX group (−4.08 ± 6.79 g/day, *p* = 0.004). With regards to fatty acid intake, no differences in intake of saturated fatty acids (SFAs) or PUFAs could be found among the study groups. In contrast, intake of monounsaturated fatty acids (MUFAs) decreased significantly in the EXCO (−3.41 ± 12.0 g/day, *p* = 0.040) and EXDC group (−4.80 ± 9.17 g/day; *p* = 0.011).

Further, the intake of EPA and DHA from the background diet did not change significantly throughout the study between the four study groups. However, the highest increase in EPA and DHA intake could be observed in the EXDC group (+0.15 ± 0.37 g/day DHA; +0.25 ± 0.33 g/day EPA).

### 3.4. Body Composition

After the intervention, a significant difference in fat mass was detected among all four study groups (*p* = 0.039) (Table 4). Fat mass (FM) decreased significantly within the EXDC group (−1.41 ± 2.13 kg, *p* = 0.008) and the EXCO group (−1.70 ± 2.45 kg, *p* < 0.001). Total body water content and lean body mass (LBM) increased significantly in CON and EXCO (CON: +0.59 ± 1.28 L, *p* = 0.015; +0.76 ± 1.74 kg, *p* = 0.018; EXCO: +0.57 ± 2.03 L *p* = 0.012, +0.79 ± 2.79 kg, *p* = 0.008). Bodyweight and BMI did not differ significantly between all study groups after the intervention.

### 3.5. Markers of Glucose Metabolism

Overall, no significant differences in any markers of glucose metabolism were detected among the study groups (Table 5). Nonetheless, the highest changes in fasting insulin and HOMA-Index was observed within the EXDC group (−2.14 ± 4.59 µE/L and −0.56 ± 1.25), followed by the EXCO group (−1.42 ± 2.86 µE/L and −0.27 ± 0.72).

### 3.6. Blood Lipids

No significant differences in blood lipids among all four study groups were detected (Table 6). However, when compared with each other, EXCO showed the greatest decrease in triglycerides at the end of the intervention (−10.5 ± 23.6 mg/dL).

## 4. Discussion

In this study, we show that 12 weeks of resistance and aerobic training combined with adherence to a healthy diet or supplementation of 2 g *Calanus finmarchicus* oil may promote body fat loss, but did not affect markers of glucose or lipid metabolism in overweight but otherwise healthy (no diagnosed chronic diseases) elderly subjects.

With regards to the EXDC group, a significant increase in vegetables, fruit, and fish intake occurred, which makes it arguable that improved diet quality supported body fat loss in this group [30]. However, it is surprising that the healthy diet did not support further changes such as an increase in LBM. Comparable studies investigating the effect of a healthy diet with a n6/n3 ratio < 2 (dietary recommendations, uncontrolled) or a n-3 PUFA rich diet (≥ 500 g fatty fish and seafood per week) combined with 2× /week resistance training in elderly women showed an increase in leg lean mass and increased hypertrophy of type 2 muscle fibers after 24 weeks [31,32]. However, it should be noted that not only the fatty acid intake of the participants was modified, but also their energy intake, macronutrient ratio, and fiber intake in the previous study. Naturally, an improved energy and macronutrient intake will potentially improve body composition. In contrast, participants from our study received general dietary recommendations with no specifications on caloric intake or macronutrient distribution. Moreover, Strandberg et al., 2015 measured body composition only in the lower extremities, whereas we used whole body measurements. The differences in estimation of body composition make it difficult to directly compare the study outcomes. Furthermore, although an increase in fish consumption could be observed in our study, no information about the type of fish (lean or fatty fish) was collected. In any case, the intakes from the food logs indicated a slightly higher intake of EPA and DHA within the EXDC group, but this was not significantly different from the other study groups.

In contrast to the EXDC group that showed only changes in FM, the EXCO group showed several changes in body composition, which may be mediated by CO. As health benefits from marine oils are commonly attributed to long-chain n-3 PUFAs, this is by far the most studied component of CO. With regard to body composition, results from a meta-analysis showed that supplementation with n3-PUFAs as monotherapy can promote moderate weight loss, including a reduction in body fat [21]. When combined with exercise some [33,34], but not all studies [35], reported an additional beneficial effect of n-3 PUFAs on body fat loss in healthy active or sedentary overweight/obese participants. However, evidence about the combined effect of n-3 PUFA supplementation and exercise on body composition in the elderly population is limited. In elderly subjects, 12 weeks of 3× per week resistance training and intake of ~14 g alpha-linolenic acid (ALA, C18:3n3) from 30 mL flaxseed oil showed no beneficial effect on body composition when compared to placebo (corn oil) [36]. Even a high dose of 3 g long-chain n-3 PUFAs (1.98 g EPA and 0.99 g DHA) did not affect body composition when combined with 3× a week resistance training over 12 weeks in male participants [37].

Contrary to findings from a recent study investigating the effect of CO and exercise in elderly women (~70 years old with a BMI of ~26.7) that showed no effect of CO on FM but only visceral fat [38], results from our study showed the highest loss of FM in the EXCO group. Unfortunately, we did not measure visceral fat, but the different outcomes can be partially explained by a slightly higher BMI in our study population. Moreover, we also found a significant increase in LBM and consequentially body water in this group. This result would support findings from earlier studies, that found beneficial effects of n-3 PUFAs on muscle protein synthesis [20,39]. However, existing data on the effects of n-3 PUFAs on muscle mass and strength is limited and conflicting [40]. More importantly, in this study the CON group also showed a significant increase in LBM and body water. Although the control group was advised to abstain from any exercise and to maintain their normal physical activity level, it is still possible that participants of this group changed their behavior due to their participation in the study. Indeed, a trend to an increase in physical activity could be observed but this trend was visible in all four study groups. Even though a validated and reliable questionnaire to report physical activity was reported, additional monitoring of physical activity via an activity tracker may have been beneficial.

Importantly, when evaluating the changes in body composition it has to be kept in mind that the study population was untrained, but cannot be classified as sedentary. This may indicate, that the intensity of the moderate exercise routine as performed in this study may not have been enough to promote changes in body composition if not supported by nutritional measures.

Findings from this study indicate that moderate exercise combined with nutritional measures had beneficial effects in reduction of FM. While changes within the EXCD group may be explained by improved diet quality [30], the effects in the EXCO group are likely driven through n-3 PUFAs in CO. When discussing the effects of n-3 PUFAs it is important to consider the bioavailability of their binding form, as this is the basic requirement for a substance to enter the body and convey its effects [41]. In CO, most n-3 PUFAs are bound as wax esters. Although it was believed that digestion and absorption of wax esters may not be as effective in mammals [42], a study by Gorreta and colleagues showed that digestion of wax esters was comparable with triglycerides and ethyl esters in rats [43]. A similar result was also found in humans when plasma concentrations of EPA and DHA were compared over a 72-h window after intake of 4 g CO (providing 260 mg EPA and 156 mg DHA) and a fish oil supplement containing ethyl esters (providing 465 mg EPA and 375 mg DHA), revealing equal bioavailability [44].

Regarding body fat loss, several potential mechanisms of n-3 PUFAs and their metabolites have been proposed, including increased fat oxidation, improved adipocyte function (i.e., increased lipolysis and reduced lipogenesis) as well as reduction of pro-inflammatory cytokines and oxidative stress in the adipose tissue [45,46,47]. Increased fat oxidation is discussed as a consequence of improved β-oxidation and biogenesis of mitochondria, which Flachs and colleagues described as a “metabolic switch” [48]. Generally, the underlying mechanisms are likely mediated by the AMP-activated protein kinase (AMPK) and involve further activation of transcription factors. With regards to the latter, particularly transcription factors from the family of peroxisome proliferator-activated receptors (PPAR), namely PPARα, PPARγ, and PPARδ were shown to play a role in adipogenesis [47]. However, such mechanistic relationships were mainly investigated in animal models or in vitro [46,49]. Therefore, it remains unclear which dose of n-3 PUFAs exerts beneficial effects on body composition in humans. In any case, when compared to previous human studies using supplements with >2 g of EPA + DHA, the daily dose of n-3 PUFAs from CO was considerably lower with only ~200 mg EPA + DHA. This amount even falls below the recommendations of the International Society for the Study of Fatty Acids and Lipids of 0.5 g EPA + DHA per day [50]. Beyond this background, other potential mechanisms or a combination of several pathways needs to be considered.

One compound present in CO that may influence body composition is astaxanthin, a carotene that also serves as a natural antioxidant [51]. A recent review discussed the potential beneficial effects of carotenes in counteracting obesity [24]. Although data in humans is limited, some studies reported a significant decrease in adipose tissue in obese subjects after carotene supplementation [52,53]. In the two available interventional studies, the dosage and composition of the carotenes widely varied. While one study used a mixture containing ~24 mg of carotenes (500 µg α-carotene, 1200 µg β-carotene, 500 µg astaxanthin, 2000 µg zeaxanthin, 10 mg lutein, 10 mg lycopene) and 10 mg γ-tocopherol that was administered twice daily [52], another study administered 9.0 mg of xanthophyll per day [53]. As those doses highly exceed the amount of astaxanthin present in 2 g of CO used in this study, the theory of combined effects of the components present in CO is further supported.

In addition to influencing body composition, exercise was also shown to improve markers of glucose metabolism and blood lipids in older individuals [54,55]. Therefore, we sought to identify potential changes in such metabolic markers in the overweight study population. Although results from our study showed no significant improvements in any metabolic markers, a trend to reduced fasting insulin was observed in the EXDC and EXCO groups. Moreover, the highest, yet not significant, reduction of triglycerides was observed within the EXCO group, which falls in line with results showing a triglyceride lowering effect of n-3 PUFAs supplementation [56]. However, whether those effects indicate further potential benefits from exercise combined with a healthy diet or CO needs to be determined in larger scale studies.

## 5. Limitations

Although BIA is an accepted tool for estimation of body composition in longitudinal studies comparing group effects [57,58,59], this method also has its limitations as it was shown to be susceptible to confounders such as vigorous physical activity, dietary intake, or hydration status [57,60,61]. However, as all measurements were conducted at the same time in the morning after an overnight fast, acute effects of activity or dietary intakes were minimized. Another limitation of our study is the use of a FFQ, which did not allow checking for consumption of fatty fish versus lean fish. Next, although a validated questionnaire to monitor the physical activity outside of the intervention was used, an additional documentation of the physical activity level via activity trackers would have been an informative addition. Future investigations on the effect of CO should also include a placebo group. A placebo was not included in this setting, as we aimed to directly compare the healthy diet versus CO instead of effects of CO only.

## 6. Conclusions

Taken together, results from this study indicate that a combination of moderate exercise with a healthy diet or CO supplementation may promote body fat loss in elderly, untrained, overweight subjects when compared to exercise only. Metabolic markers such as blood lipids and markers of glucose metabolism did not change significantly. Future research should investigate potential effects of a healthy diet (rich in n-3 PUFAs) vs. CO supplementation with and without exercise, particularly in regards to body composition.

## Figures and Tables

**Figure 1 nutrients-12-02139-f001:**
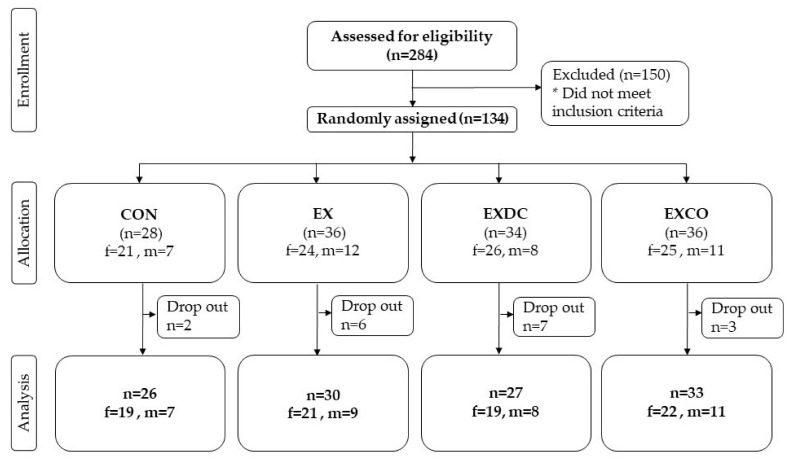
Flow chart of all participants screened, randomized, allocated and analyzed.

**Table 1 nutrients-12-02139-t001:** Composition of oil from *Calanus finmarchicus*.

Components			mg/100 g CO	mg/2 g CO
SFA			12,428	249
MUFA			11,452	229
PUFA	Omega-3		18,181	364
		ALA	1149	23
		SDA	6186	124
		EPA	5439	109
		DHA	4342	87
	Omega-6		981	20
		LA	552	11
		ARA	169	3
Fatty alcohols			33,017	660
Astaxanthin			180	3.6

CO = *Calanus finmarchicus* oil, SFA = Saturated fatty acids; MUFA = Monounsaturated fatty acids; PUFA = Polyunsaturated fatty acids; ALA = Alpha-Linolenic acid; SDA = Stearidonic acid; EPA = Eisosapentaenoic acid; DHA = Docosahexaenoic acid; LA = Linoleic acid; ARA = Arachidonic acid. Other minor compounds are triacylglycerides, free fatty acids, phospholipids and plant sterols.

**Table 2 nutrients-12-02139-t002:** Baseline characteristics of all participants.

Parameter	CON (*n* = 28)	EX (*n* = 36)	EXDC (*n* = 34)	EXCO (*n* = 36)	*p*
Sex (f/m)	21/7	24/12	26/8	25/11	0.797
Age (years)	59.3 ± 5.07	59.7 ± 6.39	60.2 ± 5.11	58.5 ± 5.69	0.563
Height (m)	169.1 ± 8.05	170.9 ± 8.73	171.0 ± 8.37	171.3 ± 8.74	0.774
Body weight (kg)	84.8 ± 22.6	81.9 ± 18.6	85.0 ± 20.3	80.7 ± 20.3	0.771
BMI (kg/m^2^)	29.5 ± 6.80	28.0 ± 5.51	28.95 ± 5.80	27.28 ± 5.26	0.427
SBP (mmHg)	138.0 ± 20.4	129.4 ± 14.7	133.8 ± 16.9	134.1 ± 20.1	0.332
DBP (mmHg)	81.1 ± 10.5	76.2 ± 6.16	79.2 ± 10.6	78.6 ± 7.91	0.327
Pulse	71.4 ± 8.57	74.6 ± 9.58	72.6 ± 9.23	73.8 ± 8.25	0.513

Values are given as mean ± SD. Distribution of sexes between groups was analyzed using Chi-square-test. All other group differences were assessed with Kruskal-Wallis-Test. f = females, m = male, BMI = Body-Mass-Index, SBP = systolic blood pressure, DBP = diastolic blood pressure.

**Table 3 nutrients-12-02139-t003:** Daily dietary intake from the background diet calculated from 3-day dietary records and absolute changes (Δ) at the beginning (0) and at the end (12) of the study.

Parameter		CON		EX		EXDC		EXCO		*p*
	t		Δ		Δ		Δ		Δ	
Energy intake (kcal)	0	1873.7 ± 354.0	−21.0 ± 51.6	2023.5 ± 408.5	−217.3 ± 455.0	1887.4 ± 539.5	−227.5 ± 453.8	1923.5 ± 436.8	−146.8 ± 476.3	0.334
	12	1852.7 ± 476.0		1806.1 ± 365.4		1659.9 ± 470.9		1776.8 ± 413.5		
Protein (%E)	0	16.7 ± 2.5	−0.08 ± 2.40	16.1 ± 3.42	0.86 ± 3.84	17.6 ± 3.66	−0.02 ± 4.05	15.6 ± 2.99	1.05 ± 3.68	0.772
	12	16.7 ± 2.0		16.9 ± 3.37		17.6 ± 2.93		16.7 ± 3.26		
Fat (%E)	0	36.1 ± 7.39	4.32 ± 8.17	36.2 ± 7.47	0.69 ± 7.97	37.2 ± 7.01	−0.60 ± 7.04	38.1 ± 6.69	−2.51 ± 9.04	0.016
	12	40.4 ± 6.24 ^†^		36.9 ± 6.00		36.6 ± 7.93		35.6 ± 7.48		
CHO (%E)	0	41.66 ± 6.23	−5.11 ± 6.65	42.6 ± 9.45	−2.76 ± 8.36	39.0 ± 6.16	0.66 ± 8.08	41.92 ± 6.09	0.47 ± 8.72	0.015
	12	36.56 ± 5.93 ^†^		39.8 ± 6.45 *		39.6 ± 7.76		42.39 ± 7.86		
Fiber (g)	0	20.80 ± 7.44	−0.76 ± 7.11	23.4 ± 7.79	−4.08 ± 6.79	19.0 ± 5.97	3.00 ± 8.39	22.0 ± 6.47	−0.78 ± 8.03	0.004
	12	20.04 ± 9.51		19.3 ± 6.08 ^†^		22.0 ± 6.43 *		21.2 ± 8.80		
SFA (g)	0	34.3 ± 14.2	2.90 ± 15.5	35.7 ± 13.6	−4.67 ± 15.7	33.8 ± 14.1	−3.61± 9.34	36.9 ± 15.5	−6.93 ± 15.2	0.060
	12	37.2 ± 13.6		31.0 ± 11.2		30.2 ±12.7		29.9 ± 11.1		
MUFA(g)	0	25.1 ± 8.53	3.51 ± 13.2	28.9 ± 12.5	−2.99 ± 11.9	27.3 ± 11.6	−4.80 ± 9.17	27.3 ± 9.61	−3.41 ± 12.0	0.030
	12	28.6 ± 11.8		26.0 ± 11.2		22.5 ± 11.5 *		23.9 ± 9.29 *		
PUFA (g)	0	10.6 ± 5.16	1.58 ± 6.83	11.9 ± 3.96	0.21± 5.01	12.4 ± 7.30	−2.13 ± 6.86	11.8 ± 5.64	−1.26 ± 4.07	0.088
	12	12.1 ± 7.69		12.1 ± 5.75		10.3 ± 5.35		10.5 ± 5.26		
DHA (g)	0	0.19 ± 0.26	−0.02 ± 0.42	0.29 ± 0.34	0.11 ± 0.46	0.26 ± 0.21	0.15 ± 0.37	0.23 ± 0.30	−0.4 ± 0.35	0.728
	12	0.18 ± 0.32		0.39 ± 0.47		0.40 ± 0.43		0.19 ± 0.19		
EPA (g)	0	0.14 ± 0.42	−0.03 ± 0.56	0.27 ± 0.54	−0.01 ± 0.37	0.12 ± 0.17	0.25 ± 0.33	0.18 ± 0.41	0.09 ± 0.23	0.095
	12	0.12 ± 0.30		0.30 ± 0.41		0.38 ± 0.42		0.14 ± 0.19		

Values are given as mean ± SD. *p* values represent time*intervention interaction analyzed with two-way repeated measure ANOVA. In case of significance, asterisks indicate statistical differences within groups detected with Bonferroni post hoc test (* *p* < 0.05; ^†^
*p* < 0.01; ^‡^). t = time in weeks, %E = percent of total energy, CHO = carbohydrates.

**Table 4 nutrients-12-02139-t004:** Body composition and absolute changes (Δ) of the four groups at the beginning (0) and at the end (12) of the study of all four study groups.

Parameter		CON		EX		EXDC		EXCO		*p*
	t		Δ		Δ		Δ		Δ	
Weight (kg)	0	84.9 ± 23.0	−0.28 ± 1.97	81.0 ± 18.5	−1.12 ± 1.84	84.8 ± 20.8	−1.26 ± 2.34	82.2 ± 20.0	−1.16 ± 2.49	0.236
	12	84.6 ± 22.2		79.9 ± 18.0		83.6 ± 20.7		81.0 ± 19.0		
BMI (kg/m^2^)	0	29.5 ± 6.93	−0.09 ± 0.69	27.8 ± 5.39	−0.38 ± 0.64	28.7 ± 5.86	−0.43 ± 0.86	27.6 ± 5.25	−0.37 ± 0.81	0.229
	12	29.4 ± 6.66		27.4 ± 5.26		28.3 ± 5.78		27.2 ± 5.01		
Phase angle	0	5.44 ± 0.77	−0.04 ± 0.26	5.47 ± 0.87	0.16 ± 0.57	5.23 ± 0.50	0.10 ± 0.30	5.51 ± 0.81	0.11 ± 0.33	0.279
	12	5.39 ± 0.83		5.61 ± 0.65		5.33 ± 0.57		5.62 ± 0.79		
TBW (L)	0	39.1 ± 7.86	0.59 ± 1.28	40.2 ± 9.17	−0.72 ± 2.71	40.4 ± 9.54	0.07 ± 1.21	40.5 ± 9.91	0.57 ± 2.03	0.028
	12	39.7 ± 7.51 *		39.4 ± 7.61		40.5 ± 9.47		41.3 ± 9.36 *		
LBM (kg)	0	53.3 ± 10.7	0.76 ± 1.74	54.8 ± 11.9	−0.87 ± 3.53	55.3 ± 13.0	0.09 ± 1.67	55.3 ± 13.5	0.79 ± 2.79	0.031
	12	54.0 ± 10.1 *		53.7 ± 10.1		55.3 ± 12.9		56.4 ± 12.6 ^†^		
BCM (kg)	0	26.3 ± 6.07	0.12 ± 1.32	27.0 ± 7.99	−0.03 ± 3.16	26.6 ± 7.24	0.35 ± 1.46	27.3 ± 7.63	0.71 ± 1.74	0.388
	12	26.4 ± 5.83		26.9 ± 6.08		27.0 ± 7.30		28.1 ± 7.18		
FM (kg)	0	31.6 ± 17.0	−0.35 ± 2.00	26.2 ± 11.2	−0.22 ± 3.03	29.6 ± 12.1	−1.41 ± 2.13	27.0 ± 11.3	−1.70 ± 2.45	0.039
	12	31.3 ± 16.5		26.2 ± 11.5		28.2 ± 12.1 ^†^		24.7 ± 11.1 ^‡^		

Values are given as mean ± SD. *p* values represent time*intervention interaction analyzed with two-way repeated measure ANOVA. In case of significance, asterisks indicate statistical differences within groups detected with Bonferroni post hoc test (* *p* < 0.05; ^†^
*p* < 0.01; ^‡^
*p* < 0.001). t = time in weeks, BMI = body mass index, TBW = total body water, LBM = lean body mass, BCM = body cell mass, FM = fat mass.

**Table 5 nutrients-12-02139-t005:** Markers of glucose metabolism and absolute changes (Δ) at the beginning (0) and at the end (12) of the intervention.

Parameter		CON		EX		EXDC		EXCO		*p*
	t		Δ		Δ		Δ		Δ	
Fasting Glucose (mg/dL)	0	92.9 ± 8.98	0.27 ± 12.46	94.0 ± 17.5	−1.40 ± 14.25	97.9 ± 26.5	+0.07 ± 10.63	88.6 ± 8.29	3.42 ± 15.97	0.668
	12	93.2 ± 14.3		92.6 ± 8.76		98.0 ± 25.5		92.1 ± 14.29		
HbA_1c_ (%)	0	5.47 ± 0.34	−0.08 ± 0.15	5.39 ± 0.28	−0.08 ± 0.17	5.58 ± 0.72	−0.05 ± 0.17	5.38 ± 0.36	−0.07 ± 0.22	0.954
	12	5.39 ± 0.33		5.30 ± 0.25		5.52 ± 0.72		5.31 ± 0.35		
HbA_1c_ (mol/molHB)	0	36.3 ± 3.75	−0.84 ± 1.65	35.4 ± 3.06	−0.90 ± 1.85	37.5 ± 7.88	−0.58 ± 1.84	35.3 ± 3.96	−0.80 ± 2.38	0.954
	12	35.4 ± 3.65		34.4 ± 2.76		36.8 ± 7.92		34.5 ± 3.79		
Insulin (µE/mL)	0	13.4 ± 11.1	−0.74 ± 4.52	9.80 ± 4.20	−0.48 ± 4.24	12.7 ± 8.48	−2.14 ± 4.59	10.2 ± 7.16	−1.42 ± 2.86	0.457
	12	12.7 ± 10.2		9.32 ± 3.88		10.5 ± 7.21		8.81 ± 5.45		
HOMA−Index	0	3.24 ± 3.19	−0.16 ± 1.19	2.40 ± 1.52	−0.23 ± 1.51	3.11 ± 2.36	−0.56 ± 1.25	2.31 ± 1.83	−0.27 ± 0.72	0.640
	12	3.08 ± 2.95		2.17 ± 1.00		2.56 ± 2.02		2.04 ± 1.41		

Values are given as mean ± SD. *p* values represent time*intervention interaction analyzed with two-way repeated measure ANOVA. In case of significance, asterisks indicate statistical differences within groups detected with Bonferroni post hoc test. t = time in weeks.

**Table 6 nutrients-12-02139-t006:** Blood lipids and absolute changes (Δ) of the four groups at the beginning (0) and at the end (12) of the intervention.

Parameter		CON		EX		EXDC		EXCO		*p*
	t		Δ		Δ		Δ		Δ	
TG (mg/dL)	0	141.9 ± 104.1	−6.46 ± 43.2	115.0 ± 55.1	−3.40 ± 33.6	113.7 ± 44.3	3.22 ± 46.5	107.3 ± 34.4	−10.5 ± 23.6	0.470
	12	135.5 ± 89.2		111.6 ± 49.7		116.9 ± 68.9		96.9 ± 28.5		
TC (mg/dL)	0	258.3 ± 54.7	−2.08 ± 33.5	237.7 ± 39.2	−5.37 ± 20.9	244.3 ± 51.6	−1.59 ± 25.8	242.2 ± 37.2	0.70 ± 25.3	0.816
	12	256.2 ± 59.6		232.4 ± 37.9		242.7 ± 49.7		242.9 ± 38.2		
HDL (mg/dL)	0	62.8 ± 14.0	−0.08 ± 6.33	67.5 ± 15.4	0.03 ± 7.45	61.1 ± 14.9	1.22 ± 6.38	68.0 ± 18.2	2.61 ± 7.54	0.410
	12	62.7 ± 14.1		67.5 ± 15.2		62.3 ± 13.8		70.6 ± 19.3		
LDL (mg/dL)	0	167.8 ± 39.6	1.00 ± 25.2	150.3 ± 31.7	−2.67 ± 14.8	157.7 ± 42.6	2.30 ± 19.19	152.0 ± 29.2	2.55 ± 18.2	0.710
	12	168.8 ± 47.3		147.6 ± 27.4		160.0 ± 40.6		154.6 ± 29.1		
LDL/HDL Ratio (mg/dL)	0	2.77 ± 0.80	0.04 ± 0.45	2.37 ± 0.88	−0.07 ± 0.31	2.75 ± 1.03	−0.07 ± 0.32	2.39 ± 0.71	−0.08 ± 0.30	0.694
	12	2.81 ± 0.98		2.31 ± 0.72		2.68 ± 0.88		2.32 ± 0.68		

Values are given as mean ± SD. *p* values represent time*intervention interaction analyzed with two-way repeated measure ANOVA. In case of significance, asterisks indicate statistical differences within groups detected with Bonferroni post hoc test. t = time; weeks of the intervention, TC = total cholesterol, TG = triglycerides.

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
