# Peer review of "Effects of Exercise Combined with a Healthy Diet or Calanus finmarchicus Oil Supplementation on Body Composition and Metabolic Markers—A Pilot Study"

_nutrients, 2020, doi:10.3390/nu12072139_

Round 1
Reviewer 1 Report
Introduction:
- Runon sentence, lines 36-38
- Incorrect sentence and grammar, lines 38-42
- replace was with has, line 41
- make clear reference to exercise training, line 42
- comma usage is incorrect, line 52 (and throughout manuscript)
- promising--type line 55
- PUFA should be introduced in line 50 instead of 59
- mention of bone, line 37, is irrelevant since not focus of work
- while other effects of other components of CO should be mentioned, many points in the discussion, lines 373-382, should be moved to the introduction
- line 68, how is this supplementation with a single nutrient? (table 1)
- Overall, introduction needs to more clearly layout purpose of study and why specific groups were designed.
- participants are overweight--this is not mentioned in purpose
Methods:
- missing CO only group needs to be justified
- why not look at omega-3 index to ensure CO compliance
Results:
- Gaps in lines in table 3 seem odd
- Overall, graphs in place of tables would be better
- missing bottom line of table 4?
Discussion
- body composition method used should be mentioned as a limitation
- pre-obese?
- participants referred to as elderly but fact that they are overweight is worth mentioning
- discussion should be reorganized to include a limitations paragraph--much of which the discussion should go under
Author Response
Point-by-point response to the comments made by the reviewers
We thank the reviewers for their kind comments and the opportunity to revise our paper. The comments and suggestions have helped to improve the manuscript substantially. In the following, each comment is addressed separately. All answers to the comments are highlighted in blue, while changes in the revised manuscript are highlighted in red.
Reviewer 1
Introduction:
1.Runon sentence, lines 36-38
The sentence was now adjusted. Please see l. 36-39 for reference.
2.Incorrect sentence and grammar, lines 38-42
We corrected this section. Please see l. 38-40 for reference.
3. replace was with has, line 41
This sentence was now rewritten. Please see l. 39-41.
4.make clear reference to exercise training, line 42
As this section was adjusted, the refences are now more clearly linked to exercise. We also added another meta-analysis as reference (Nocon et al. 2008). Please see l. 39-41 for the changes made.
5. comma usage is incorrect, line 52 (and throughout manuscript)
Following your recommendation, the manuscript was proofread by a native speaker.
6.promising--type line 55
The typo was corrected (now l. 57).
7.PUFA should be introduced in line 50 instead of 59
We now changed n-3 FAs to n-3 PUFAs throughout the manuscript. The abbreviations is introduced in l. 53.
8.mention of bone, line 37, is irrelevant since not focus of work
We agree and deleted the information about loss of bone mass.
9.while other effects of other components of CO should be mentioned, many points in the discussion, lines 373-382, should be moved to the introduction
Our intention is to balance the background information between the three intervention groups within the introduction. Hence, we did not want to include too much information about single components of CO. However, according to your suggestions, we rewrote major parts of the introduction and included information about potential effects of carotenoids on obesity (please see l 62-63 for reference).
10.line 68, how is this supplementation with a single nutrient? (table 1)
As the introduction was restructured, we hope we could clarify why we mention supplementation of individual nutrients (l. 51-55).
11.Overall, introduction needs to more clearly layout purpose of study and why specific groups were designed.
Thank you for this hint. We made major changes to the introduction to clarify the purpose of each study group. Please see all changes marked in red throughout our introduction (l.36-75).
12.participants are overweight--this is not mentioned in purpose
Indeed, this information is not mentioned in the purpose. This is explained by the fact that we did not primarily aimed to recruit an overweight study population (e.g. no minimum BMI of 25 kg/m2 was set as prerequisite for participation in the study).
Methods:
1.missing CO only group needs to be justified
We agree, that CO only would have been a valuable addition to our study. However, our study chose to investigate effects of exercise versus exercise + a healthy diet versus exercise + CO. The inclusion of a “CO only” group would not have fit into our primary research question. Nonetheless we understand your point and added some information on this in the conclusion as a outlook for future studies (l. 410-412).
2.why not look at omega-3 index to ensure CO compliance
In fact, an analysis of the omega-3 index was performed in the EXCO group, as well as in the EX group as a control (to account for potential effects of exercise). A paper including all detailed results on the omega-3 index is currently under review elsewhere.
Results:
1.Gaps in lines in table 3 seem odd
Thank you for this hint, we deleted gaps found in the last row of table 3.
2. Overall, graphs in place of tables would be better
We agree, that visualization of data in figures is always great. However, in this case we believe that the data can be presented more clearly in tables.
3. missing bottom line of table 4?
We now added the missing bottom line.
Discussion
1.body composition method used should be mentioned as a limitation
We now included a limitations section. Please see l. 394-405 for reference.
2.pre-obese?
Following your recommendation, we now changed this word to overweight throughout the manuscript.
3.participants referred to as elderly but fact that they are overweight is worth mentioning
Thank you for pointing this out. We now included that participants were overweight (see l. 295-296 for reference). We also clarified, that “healthy” means that participants had no diagnosed chronic disease.
4.discussion should be reorganized to include a limitations paragraph--much of which the discussion should go under
We followed your recommendation and addressed limitations in a separate limitations section. Please see l. 394-405 for reference.

Reviewer 2 Report
In the study by Wasserfurth et al. the authors raise the question of whether exercise or exercise combined with nutritional interventions can counteract the commonly observed decline in muscle mass and increase in fat mass in untrained elderly people. Thus, they performed a randomized 12 wk controlled interventional trial, including 134 healthy, untrained women and men, which were subdivided in four study groups: (1) control group with no intervention (CON); (2) 2x/week aerobic and resistance training only (EX); (3) exercise routine combined with dietary counseling (EXDC); (4) exercise routine combined with intake of 2 g/day oil from Calanus finmarchicus (EXCO).
Analysis of body composition at the beginning and the end of the study showed a significant decrease in body fat in the EXDC and EXCO groups (1.41± 2.13 and 1.70± 2.45 kg, respectively), while markers of glucose metabolism and blood lipids remained unchanged in all groups. The authors conclude that a combination of moderate exercise and intake of oil from Calanus finmarchicus or a healthy diet may promote fat loss in elderly untrained, pre-obese participants.
The manuscript is very well written and easy to read. Furthermore, the study seems well controlled, in particular the recruitment and screening of the participants for eligibility. However, the study would have benefitted from inclusion of two more groups, namely participants receiving dietary counseling or oil from Calanus finmarchicus without combination with exercise. This would have allowed the authors to determine if the observed anti-obesogenic effect of either dietary counseling or oil from Calanus finmarchicus is dependent on exercise or not. If it is so that the relatively low level of exercise used in the current study can “catalyze” the effect of the treatments, it would be a very strong incentive for pre-obese subjects to combine exercise with a healthy diet or dietary supplementation with marine oil in order to reduce body fat. This referee is aware, however, of the complexity it would impose by including two more groups (especially the statistical evaluation of the results), but would strongly recommend the authors to follow up in future experiments the question of whether the fat-reducing effects of e.g. oil from Calanus finmarchicus is dependent on exercise or not.
There is no statistical evidence for the statement in the abstract that “the highest decrease in body fat was observed within the EXCO group”. The sentence should therefore be modified. On the other hand, only the EXCO group showed increases total body water and lean body mass along with the reduction in body fat, demonstrating that the effect of oil from Calanus finmarchicus is robust.
Author Response
Point-by-point response to the comments made by the reviewers
We thank the reviewers for their kind comments and the opportunity to revise our paper. The comments and suggestions have helped to improve the manuscript substantially. In the following, each comment is addressed separately. All answers to the comments are highlighted in blue, while changes in the revised manuscript are highlighted in red.
Reviewer 2
In the study by Wasserfurth et al. the authors raise the question of whether exercise or exercise combined with nutritional interventions can counteract the commonly observed decline in muscle mass and increase in fat mass in untrained elderly people. Thus, they performed a randomized 12 wk controlled interventional trial, including 134 healthy, untrained women and men, which were subdivided in four study groups: (1) control group with no intervention (CON); (2) 2x/week aerobic and resistance training only (EX); (3) exercise routine combined with dietary counseling (EXDC); (4) exercise routine combined with intake of 2 g/day oil from Calanus finmarchicus (EXCO).
Analysis of body composition at the beginning and the end of the study showed a significant decrease in body fat in the EXDC and EXCO groups (1.41± 2.13 and 1.70± 2.45 kg, respectively), while markers of glucose metabolism and blood lipids remained unchanged in all groups. The authors conclude that a combination of moderate exercise and intake of oil from Calanus finmarchicus or a healthy diet may promote fat loss in elderly untrained, pre-obese participants.
The manuscript is very well written and easy to read. Furthermore, the study seems well controlled, in particular the recruitment and screening of the participants for eligibility. However, the study would have benefitted from inclusion of two more groups, namely participants receiving dietary counseling or oil from Calanus finmarchicus without combination with exercise. This would have allowed the authors to determine if the observed anti-obesogenic effect of either dietary counseling or oil from Calanus finmarchicus is dependent on exercise or not. If it is so that the relatively low level of exercise used in the current study can “catalyze” the effect of the treatments, it would be a very strong incentive for pre-obese subjects to combine exercise with a healthy diet or dietary supplementation with marine oil in order to reduce body fat. This referee is aware, however, of the complexity it would impose by including two more groups (especially the statistical evaluation of the results), but would strongly recommend the authors to follow up in future experiments the question of whether the fat-reducing effects of e.g. oil from Calanus finmarchicus is dependent on exercise or not.
We thank you for this evaluation of our study and the thoughts on a study design which would have allowed to control for an effect without exercise. We agree, that this information would be very valuable in evaluating the anti-obesogenic effects. Anyway, we also believe that this is something that needs to be investigated in future studies. We included this information in in the conclusion (see l. 411-412 for reference).
There is no statistical evidence for the statement in the abstract that “the highest decrease in body fat was observed within the EXCO group”. The sentence should therefore be modified. On the other hand, only the EXCO group showed increases total body water and lean body mass along with the reduction in body fat, demonstrating that the effect of oil from Calanus finmarchicus is robust.
We changed the wording in the abstract. Please see l. 28-29 for reference
